# Evaluation of the Thermal Response of the Horns in Dairy Cattle

**DOI:** 10.3390/ani13030500

**Published:** 2023-01-31

**Authors:** Marijke Algra, Lara de Keijzer, Saskia S. Arndt, Frank J. C. M. van Eerdenburg, Vivian C. Goerlich

**Affiliations:** 1Population Health Sciences, Animals in Science and Society, Faculty of Veterinary Medicine, Utrecht University, 3584 CS Utrecht, The Netherlands; 2Population Health Sciences, Farm Animal Health, Faculty of Veterinary Medicine, Utrecht University, 3584 CS Utrecht, The Netherlands

**Keywords:** mutilation, farm animal management, bovine, welfare, adaptation, thermal window, infrared thermography

## Abstract

**Simple Summary:**

The horns of dairy cows are typically removed at a young age. However, surprisingly little is known about the biological function and the role of horns in the regulation of body temperature, or about the potential effects of horn removal for the cow. Farmers reported that horns get warmer during rumination, and studies on goats indicate that horns help with the regulation of body temperature. To study the possible function of horns in the regulation of body temperature in dairy cows, we used infrared thermography to measure the superficial temperature of the horns, eyes, and ears of 18 cows on three different farms in the Netherlands. Social and non-social behaviours of these cows were registered as well. Based on environmental temperature, humidity, and wind speed, the heat load index (HLI) was calculated as a measure of the heat load experienced by a cow. The temperature of the horns rose by 0.18 °C per unit HLI, indicating that the horns serve to lose heat. Dehorned cows had higher eye temperatures than horned cows, though this result may not be reliable due to the low sample size and experimental setup. We did not, however, find changes in horn temperature during rumination, nor with any other behaviours. Our study supports a role of horns in the regulation of body temperature, but not related to rumination. These results should be considered when measuring the potential effects of horn removal, a painful procedure.

**Abstract:**

Dairy cattle are typically disbudded or dehorned. Little is known, however, about the biological function and role of horns during thermoregulatory processes in cattle, and thus about the potential physiological consequences of horn removal. Anecdotal evidence suggests that dairy cow horns increase in temperature during rumination, and few studies on other bovid species indicate that horns aid thermoregulation. The objective of this study was, therefore, to elucidate a possible thermoregulatory function of the horns in dairy cattle. Using non-invasive infrared thermography, we measured the superficial temperature of the horns, eyes, and ears of 18 focal cows on three different farms in a temperate climate zone under various environmental circumstances. Observations of social and non-social behaviours were conducted as well. Based on environmental temperature, humidity, and wind speed, the heat load index (HLI) was calculated as a measure of the heat load experienced by a cow. The temperature of the horns increased by 0.18 °C per unit HLI, indicating that horns serve the dissipation of heat. Dehorned cows had higher eye temperatures than horned cows, though this result should be interpreted with caution as the low sample size and experimental setup prevent casual conclusions. We did not, however, find changes in horn temperature during rumination, nor with any other behaviours. Our study thus supports a role of horns in thermoregulation, but not related to rumination. These results should be considered when assessing the potential consequences of horn removal, a painful procedure.

## 1. Introduction

In most cattle breeds, both males and females grow horns. Disbudding and dehorning are considered routine practices in more than 80% of dairy cattle in the European Union [1,2]. Disbudding refers to the removal of the horn buds in young calves, whereas dehorning is an amputation of the horns of older calves/cows when disbudding is no longer an option [3]. Several reasons are associated with these practices. For example, horned cattle are perceived to be more aggressive and there is a higher risk of causing injuries to herd mates and handlers [4,5]. Removing the horns of cattle minimizes the husbandry space necessary for, e.g., lying, feeding, and drinking [4,5,6]. Moreover, current barn systems are equipped with feed racks and other installations where cows may get stuck with their horns, leading to painful damage (personal observation, Figure 1). Thus, the disbudding/dehorning of cows is said to increase the welfare of the animals by decreasing the risk of injury (either to themselves, other cows, or the farmers) and is performed due to economic considerations (e.g., reduced housing space requirements), e.g., [4]. On the other hand, however, pain resulting from disbudding and dehorning has been associated with physiological (e.g., increased cortisol levels) and behavioural changes (e.g., vocalizations, head rubbing) [3,7,8,9,10,11], and found to induce negative emotional states, indicated by ‘pessimistic’ bias [12], very likely reflecting poorer welfare, e.g., [13].

In bovids, horns have several functions. In males, during intrasexual competition for mates, horns are either used as a weapon to attack the opponent, or as a defensive shield [14,15]. Females use their horns to defend themselves and their offspring and, like males, as weaponry in intrasexual competition for resources [14,16,17]. Horns can also be used for self-grooming by scratching body parts that are difficult to reach [18,19]. Bovine horns consist of a hollow bony core that is projected from the frontal bone, covered by a keratin sheath [20]. After approximately six months of age, the bony core is pneumatized from the caudal frontal sinus. The bony core of the horn, as visible in Figure 1, is supplied with blood from the cornual artery that stems from the superficial temporal artery [21]. The horns are innervated by the cornual branch of the zygomaticotemporal nerve, a division of the trigeminal nerve, the supraorbital nerve, and the infratrochlear nerve [20,21,22]. The anatomy of bovid horns matches the description of thermoregulatory organs [5,23]. Evidence from goats suggests that the circulation of blood in the horns influences brain temperature [24]. Cows, as well as giraffes, waterbucks, sheep, and goats, use a process called nasal heat exchange to cool down exhaled air and thus reduce water and heat loss [25,26]. Later, Irrgang [27] suggested that because the hollow core of the horn is connected to the frontal sinus, the horns might participate in this heat exchange. Functionally, horns might participate in heat exchange by preserving or dissipating heat through vasomotor changes in the superficial blood vessels. This physiological process can be visualized and quantified by infrared thermography [28].

Additionally, Picard et al. [29] found that bovids from temperate environments had a reduced core-to-sheath ratio and a thicker keratin sheath compared to bovids from tropical climates. These differences in horn morphology may be adaptations to either facilitate or restrict heat/water loss. After all, warmer climates are often also dryer, increasing the need for water preservation. Another indication of a thermoregulatory function of horns is the finding that breeds of polled cattle (cattle born without horns) tend to have larger ears as a proposed compensation mechanism [30]. Moreover, it has been shown that avian bills have a function in thermoregulation [31]. As analogies can be found between avian bills and bovine horns (both are highly vascularised and poorly insulated), this might indicate that bovine horns also have a thermoregulatory function. Further, anecdotal, evidence comes from farmers keeping horned cows. The farmers report horns to be warmer when the cow is ruminating [32]. Possibly, the horns aid in the dissipation of heat that is produced by the active chewing muscles during rumination.

If horns indeed serve thermoregulation, removing horns will restrict cows in their ability to adapt to changes in environmental conditions. Notably, the ability to adapt is crucial for an animal’s welfare [2,13,33].

The present study was initiated to elucidate a possible thermoregulatory function of the horns in dairy cattle. Moreover, since farmers reported cows to have warmer horns during rumination, this aspect was investigated as well. Temperature measurements were performed using infrared thermography, a non-invasive tool to measure superficial body temperature in cattle [34]. We expected horn temperature to positively correlate with environmental parameters, such as temperature, humidity, and heat load index, and negatively with wind speed. As it was conceivable that horns aid the dissipation of heat created during rumination, we expected horn temperature to rise during a rumination bout.

## 2. Materials and Methods

### 2.1. Ethical Statement

The cows were only observed and not touched or interfered with. Hence, an animal experimentation license was not required according to the European Directive 2010/63/EU and the Dutch Experiments on Animals Act (WOD) as amended on 18 December 2014.

### 2.2. General Study Set-Up

The first part of the study aimed to determine whether horn temperatures differ between different behavioural and environmental circumstances. The methods involved behavioural observations and infrared thermography measurements (IRT) at three dairy farms in the Netherlands. Two farms kept horned cows; the other farm kept cows whose horns were removed. In total, 18 cows were followed from April 2021 until July 2021. IRT images of horns, eyes, and ears were collected simultaneously with behavioural observations. Several environmental parameters (air temperature, humidity, and wind speed) were measured at the beginning of each observation. The second part of the study investigated the hypothesis that horn temperature increases during rumination. In total, 42 cows were followed between July and October 2021. IRT images were taken repeatedly during entire rumination bouts at the same two dairy farms keeping horned cows.

The methodological procedures for the data collection and an ethogram were developed based on a review of the relevant literature [34,35,36,37,38,39,40,41,42,43,44,45,46,47,48,49,50,51]. The methods were then tested and refined during a pilot study of two weeks at ‘De Tolakker’, the research farm of the Faculty of Veterinary Medicine of Utrecht University. Prior to data collection, the observers (biology and veterinary medicine MSc students) were trained by staff on cattle behaviour and IRT measurements. During two weeks prior to the start of the data collection, the IRT protocol and ethogram were further tested and validated on the participating farms of this study. These preliminary data were not used in the final analysis.

### 2.3. Focal Animals

For the first part of the study, six cows on each of the three farms were selected as focal individuals. The cows were selected by first excluding all individuals that were in the non-desired lactation stages during the observation period. As the metabolism of cows changes drastically in the transition period, e.g., [52], it was decided to exclude females that calved less than 8 weeks before observations started or that were expected to calve less than 6 weeks after the observations ended. However, due to miscalculations, there were three cows (one from each farm) in the selection that were in their dry period for (part of) the observation period. Cows with horns inequal in size (due to fractures) were excluded as well. The remaining cows were then divided into three age categories. The oldest category consisted of individuals born in 2014 or earlier, the middle category of individuals born in 2015 or 2016, and the young category of cows born in 2017 or later, that gave birth to at least one calf. Two cows per age category were selected randomly using R software (R version 4.0.3, 10 October 2020). Additionally, for farm A, the cows were divided based on their coat colour (black or red), so that for each colour and age combination only one cow was selected. An overview of the breed characteristics and images of the focal cows can be found in Appendix A.

For the second part of the study, cows from farms A and B were excluded based on the same criteria (lactation stage and unequal horn size). This selection resulted in 11 focal cows from farm A and 31 focal cows from farm B.

#### 2.3.1. Farm A

Farm A kept approximately 25 adult horned cows in a loose housing system. This farm kept several breeds of cattle, including Holstein-Friesian, Blaarkop (an old Dutch dual-purpose breed), German Black Pied, Meuse-Rhine-Yssel, one Jersey, and some mixed breeds (Appendix A). The average yearly milk yield of the focal cows was 5230 kg (range: 4474–5924 kg). The barn was equipped with 110 m^2^ of straw bedding and 110 m^2^ of slatted floors with feeding troughs. However, for most observation days (except for one day, due to a snowstorm), the cows had access to the outdoor pasture. The average environmental temperature on the observation days was 16 °C and the average humidity 46%. The size of the pasture varied between 5500 and 41,000 m^2^ and cows grazed several different fields across the days. The fields were separated by small canals, which offered ad libitum water supply. In the warmer months (from March until November), the cows mostly fed on grass and some hay or grass silage when they were in the barn for milking. From November until March, the cows were housed indoors and received hay and concentrated feed. From April 13th onwards, the cows were generally left outside overnight as well. Cows were only inside for milking twice a day, at 8:30 and 17:30. On most of the more rainy and/or warm days, cows had access to the barn so that they could choose to stay inside or go outside. The dry cows were always on pasture but did not have the option to go into the barn. Lastly, inside the barn, the cows had ad libitum access to an automated cattle brush.

#### 2.3.2. Farm B

Farm B kept approximately 90 adult horned Jersey cows (Appendix A). The average yearly milk yield of the focal cows was 5085 kg (range: 4124–5758 kg). The cows were always on pasture when observations were performed. The average environmental temperature on the observation days was 18 °C and the average humidity 45%. The field size differed each week, but was generally between 20,000 m^2^ and 75,000 m^2^. In the field, cows had ad libitum access to a water trough and to a container with a mixture of herbs. The cows were milked each day at 6:00 and again at 17:00. The dry cows were generally housed indoors, in a part of the barn of 300 m^2^ where they had ad libitum access to hay and/or silage and water. In the warmer months (March until November), the cows were fed 100% grass; during winter, they received extra wheat. At the edges of the field were some trees where the cows could find shelter when it was raining or very warm. At this farm, the cows were not inseminated until they were 100 days post-partum. Inside the barn the cows had ad libitum access to an automated cattle brush.

#### 2.3.3. Farm C

Farm C housed approximately 85 adult Holstein-Friesian cows (Appendix A) in a housing system of 650 m^2^, with slatted floors and cubicles. All cows were disbudded at young age. The average yearly milk yield of the focal cows was 8171 kg (range: 6417–10,072). The cows were milked twice a day, at 06:00 and at 18:00. The dry cows were separated from the main group, in a closed-off section of the barn, and always housed indoors. From the 2nd of April, the first observation day, to the 23rd of April, the cows were always indoors. From the 23rd of April onwards, except for the 28th of May due to bad weather, the cows were left outside at around 09:00, and were inside at around 17:30. The cows stayed indoors overnight. The average environmental temperature on the observation days was 16 °C and the average humidity 55%. The sizes of the pasture fields varied from 11,415 m^2^ to 56,227 m^2^. Water troughs were available in several areas of the barn, and the fields offered access to a water trough or canal. When the cows were housed indoors, grass silage, mineral salt, corn silage, water, and an automated cattle brush were available ad libitum. Concentrated feed was available through an automated dispenser.

### 2.4. Horn Temperature in Relation to Behaviour and Environmental Conditions

#### 2.4.1. Behavioural Observations

For the first part of the study, data were collected from the end of March to July 2021 by two observers. Before March 30th, all three farms were already visited twice, so the observers could get accustomed to the cows, the farm, and the equipment. These days were not used in the data analysis.

During data collection, each farm was visited once a week. Each data collection day, 30 min were spent in proximity of the cows prior to starting observations, to let the cows get accustomed to the presence of the observers. The aim was to perform five rounds of five-minute observations of each of the six focal cows, between 9:00 and 17:00, to avoid milking times. Due to weather or other unforeseen circumstances it was not always possible to complete all five rounds on a day; in that case, additional observations were performed on following days. The order in which the cows were observed was randomized for each week separately. In total, 50 rounds were performed, resulting in a total of 250 min of recorded data per cow (Table 1).

Each observation lasted for five minutes after a 30 s introduction. To record the behaviours, a camcorder (HC-V180 Camcorder, Panasonic, the Netherlands) was used. The introduction time was used to record the date, time, air temperature, humidity, wind speed, wind direction (as seen from the cow), cow number, and location. The environmental parameters (air temperature, humidity, and wind speed) were measured using a Kestrel 3000 (Kestrel, Boothwyn, PA, USA). After the first 30 s of video recording, the observer verbally mentioned every behaviour that the cow was performing using the focal animal sampling method [53]. Behaviour was characterized based on a comprehensive ethogram. See Appendix A, for the detailed protocol of the behavioural observations. For the statistical analyses in relation to horn temperature, a subset of behaviours was selected (Table 2). Because the behaviours “drinking”, “being alert”, “milking”, and “no non-social behaviours” only occurred during 10 rounds, they were grouped into a category “other”.

#### 2.4.2. Infrared Temperature Measurements

During the five-minute observation bouts, the second observer handled the thermal imaging camera (FLIR T530 IR camera, FLIR Systems Inc., thermalfocus, Belgium) and took as many IRT images as possible. Radiometric images were taken of the eye, ear, and horn (Appendix A), alternating sides when possible. The images were taken at an angle between 90° and 45° (ensuring the eye and ear were visible from the front) and at a distance to the head of the subject of around 1 to 1.5 m [37,43,54,55,56,57,58]. The resolution of the images was 640 × 480. Images of the horns were excluded from analysis if the base of the horn was not visible. To analyse changes in horn temperature during rumination, the radiometric images were taken frontally. See Appendix A, for the detailed protocol of the thermal image collection.

### 2.5. Horn Temperature in Relation to Rumination

The second part of the study aimed at investigating whether horn temperature changed during rumination. A third observer collected IRT data on farm A and farm B (horned cattle) twice a week between July and October 2021. All the observations were carried out between 09:00 and 17:00, avoiding milking times. Within this time frame, the start of a trial was random, depending on when a cow commenced rumination. When a cow lay down, she was closely observed for the start of rumination. From the moment the cow started ruminating, IRT images were taken with a thermal imaging camera (FLIR T430sc IR camera, FLIR Systems Inc., KWx, Belgium). The IRT images were taken at approximately 1 m distance and the camera was directed frontally to the horns (Appendix A). The images were taken at an interval of one minute for the first ten minutes, then every two minutes for another ten minutes, then every five minutes for the rest of the rumination bout. At the start of rumination, the environmental temperature, relative humidity, and wind speed were measured with the same Kestrel 3000 as used previously. These measurements were repeated after every 20 min of rumination. The Kestrel was held at the level of the lying cow’s head and as far from the observer’s head as possible, to prevent the exhaled breath from interacting with the sensors of the device. The cows were always photographed in the pasture and for as long as they were ruminating in a lying position.

### 2.6. Data Processing

For the first part of the study, the observer that did not film the behaviours during the observations scored the videos according to the ethogram (Table 2). The observer that did not take the IRT images analysed the regions of interest, thus the temperature of horns, eyes, and ears.

The behavioural recordings were scored with Solomon Coder (version: beta 19.08.02, available online at https://solomon.andraspeter.com/, accessed on 30 October 2020). For each recording, the body position, wind direction (as seen from the cow), location, social circumstance, and behaviours (state and event) were coded. Only state behaviours were used for the current analyses, as these were expected to potentially affect horn temperature. The duration of state behaviours was registered when initiated until the behaviour was interrupted by another behaviour or stopped for more than 10 s.

The thermal images were analysed using a Noldus laptop with FLIR’s ResearchIR Max (version: 4.30.1.70 64-bit (9 March 2016)). From each observation, only sixteen images were selected to avoid bias from easier to observe behaviours. The highest quality images were selected based on the criteria visibility (the three regions of interest (ROIs), i.e., horns, ears, and eyes, should be completely visible), sharpness, and correct angle. If multiple images were taken within 10 s, only the image with the highest quality according to the aforementioned criteria was used. For each image, the visible ROIs were selected using the ROI tool of the FLIR software. Specifically, the horn ROI was selected using the polygon function and following the perimeter of the horn, without including the hairs at the base and making sure the warmest spot was encompassed. For each ROI, the maximum temperature was then determined by the FLIR software and saved for analysis, in accordance with previous publications [42,43]. Regarding image parameters, emissivity was set at 0.98 and the reflected temperature at 20 degrees Celsius, in accordance with previous reports [43,56]. See Appendix A for the detailed protocol of the data processing.

The behavioural data were then combined with the IRT data using R studio [59,60]. An R script was used that identified which duration behaviours were occurring at the time of each IRT photograph (Appendix A).

For the second part of the study, all images and measurements were collected and analysed by the third observer. The obtained IRT images were analysed based on the same method as in the first study, except that all photographs per observation were included.

#### HLI and THI Calculation

To investigate horn temperature in relation to environmental parameters, the heat load index (*HLI*) was calculated using the following formulas [61]:(1)$$HLI_{BG>25}=8.62+\left(0.38\times RH\right)+\left(1.55\times BG\right)-\left(0.5\times WS\right)+{\;e}^{\left(2.4-WS\right)}$$
(2)$$HLI_{BG<25}=10.66+\left(0.28\times RH\right)+\left(1.3\times BG\right)-WS$$

In these formulas, *RH* stands for the relative humidity (%), *WS* is the wind speed (m/s), and *BG* is the black globe temperature, which can be calculated using the following formula [62]:(3)$$Predicted\;BGT=1.33\times T-2.65\times T^{0.5}+3.21\times log_{10}\left(SR+1\right)+3.5$$

In this formula, *T* stands for the air temperature (°C) and *SR* is the solar radiation (W/m^2^).

For the solar radiation, the global radiation (*Q*), in joule/cm^2^, was obtained for each observation day from the Royal Netherlands Meteorological Institute [63]. The weather station closest to farm A was 22 km away, the weather station closest to farm B was 31 km away, and the weather station closest to farm C was 4.6 km away. Global radiation was converted to solar radiation with the following equation:(4)SR=Q×10,000/86,400

To investigate changes in horn temperature during rumination, environmental temperature and relative humidity were combined to a temperature–humidity index (*THI*) score based on the following formula [64,65,66]:(5)$$THI=1.8\times T-\left(1-RH\right)\times \left(T-14.3\right)+32$$

### 2.7. Statistical Analysis

Data were analysed using R studio [59,60] and the following packages: ‘dplyr’, ‘EnvStats’, ‘FSA’, ‘gridExtra’, ‘lme4′, ‘doBy’, ‘ggplot2′, ‘ggpubr’, ‘grid’, ‘pastecs’, ‘readxl’, and ‘rmcorr’ [67,68,69,70,71,72,73,74,75,76,77]. The script can be found in Appendix A. The significance threshold was set at *p* < 0.05. For all tests where normality is assumed, normality was checked using the Shapiro–Wilk test and histograms. Generally, parametric tests were used if all assumptions were met. However, for the comparison of eye and ear temperatures between farms, the non-parametric alternative of the statistical test was chosen to ensure consistency between the analyses.

#### 2.7.1. Inter- and Intra-Observer Reliability

Intra- and inter-observer reliability between the two observers was calculated three times: at the beginning of the first batch, at the end of the first batch, and at the beginning of the second batch (Appendix A).

The reliability of the scoring of the IRT images was calculated using an intraclass correlation coefficient (ICC, ‘irr’ package (version 0.84.1), Appendix A). The ICCs were then interpreted following Cicchetti [78], valuing an ICC smaller than 0.4 as ‘poor’, an ICC between 0.4 and 0.59 as ‘fair’, an ICC between 0.6 and 0.74 as ‘good’, and an ICC larger than 0.75 as ‘excellent’.

The reliability of the scoring of the behaviour data was calculated in Excel using Cohen’s kappa (k) and the following formula: k = (P_o_ − P_c_)/(1 − P_c_), where P_o_ is the proportion of agreement and P_c_ is the chance of agreement (see Appendix A for the exact calculations). Using the guidelines of Landis and Koch [79], a kappa smaller than 0.00 was interpreted as ‘poor’, between 0.00 and 0.20 as ‘slight’, between 0.21 and 0.40 as ‘fair’, between 0.41 and 0.60 as ‘moderate’, between 0.61 and 0.80 as ‘substantial’, and between 0.81 and 1.00 as ‘almost perfect’.

#### 2.7.2. Horn Temperature in Relation to Behaviour and Environmental Conditions

The R package ‘rmcorr’ was used to carry out repeated measures correlations (RMCs) between horn temperature and several individual and combined environmental parameters [69]. This method was chosen over a simple correlation test to avoid pseudoreplication due to repeated measurements on the same individuals. Since the environmental parameters were only measured once per observation of 5 min and there were multiple pictures taken within the observation, the horn temperatures were averaged per observation. RMC assumes a linear relationship between the variables, which was verified using a scatterplot.

To see if there were differences in horn, eye, and ear temperatures, a Wilcoxon rank sum test (for horn temperature, since only farms A and B had horned cows so only two groups were compared) and Kruskal–Wallis tests (for ear and eye temperatures) were performed. To correct for differences between individuals, data were averaged per round. If the Kruskal–Wallis test was significant, Dunn’s test was used for post hoc comparisons between the groups. The potential difference in HLI between farms and the horn temperatures during state behaviours were tested using a Kruskal–Wallis test as well.

#### 2.7.3. Horn Temperature in Relation to Rumination

The mean duration of a rumination bout was 31 ± 10 min (*n* = 42 cows). Because the shortest rumination bout lasted 20 min, only the first 20 min of all rumination observations were analysed. The temperature values (from t = 1) were standardized for the starting temperature (t = 0). For each cow, the starting temperature was set to ‘zero’ and the initial horn temperature was subtracted from the horn temperature measured at every subsequent time point.

Horn temperature was analysed using a linear mixed-effects model, including cow ID as random effect and the following fixed effects: time, farm, windspeed, THI score and parity. Windspeed, THI score, and parity were handled as categorial variables based on the quartiles. The variables time and farm were retained in the final model regardless of the significance, because time was the variable of interest and the variable farm to control for differences in farm-related factors. The model residuals were plotted against the predicted horn temperature, showing sufficient linearity, and the QQ plots showed an acceptable normal distribution. One cow was excluded from the linear mixed model due to camera dysfunction in the first 20 min and therefore missing data.

## 3. Results

### 3.1. Inter- and Intra-Observer Reliability

For the intra-observer reliability of both observers for farms A and C, and for observer 2 for farm B, the intraclass correlation coefficient (ICC) was ‘excellent’. The inter-observer reliability was also ‘excellent’ in all measurements. See Appendix A for the coefficients, their confidence intervals, F-value, *p*-value, and the percentage of images that were included or excluded after agreement between both observers.

### 3.2. Horn Temperature in Relation to Behaviour and Environmental Conditions

The number of pictures of the left or right side of the cows were comparable (left: 4215, right: 4161; for the counts per cow see Appendix A). Cow B2 lost the keratin sheath of her horn, so the data after the incident were excluded (12th of May, 7 out of 12 days). A summary of the minimum, maximum, and mean of the parameters is given in Table 3.

The HLI did not significantly differ between the measurements at the three farms (Kruskal–Wallis test, Chi squared = 4.906, df = 2, *p* = 0.086). Therefore, ear and eye temperatures could be compared between the farms. An overview of the ROI temperature measurements for the three farms is shown in Figure 2. The horn temperature did not significantly differ between farms A and B (Wilcoxon rank sum test, W = 1154, *p* = 0.5). The ear temperature did not differ significantly between the three farms either (Kruskal–Wallis test, Chi squared = 3, df = 2, *p* = 0.3). Eye temperature differed significantly between the three farms (Kruskal–Wallis test, Chi squared = 39, df = 2, *p* < 0.001), and a post hoc Dunn’s test revealed that eye temperature was significantly higher on farm C compared to farm A (*p* < 0.001) and farm B (*p* < 0.001).

Horn temperature (averaged per observation) was positively correlated to HLI and air temperature, whereas humidity and wind speed showed a negative correlation (Figure 3). All repeated measures correlations were significant (HLI: *r_rm_*(560) = 0.60, 95% CI [0.54, 0.65], *p* < 0.001, environmental temperature: *r_rm_*(560) = 0.70, 95% CI [0.65, 0.74], *p* < 0.001), humidity: *r_rm_*(560) = −0.30, 95% CI [−0.38, −0.23], *p* < 0.001), wind speed: *r_rm_*(560) = −0.20, 95% CI [−0.28, −0.12], *p* < 0.001).

Horn temperature did not differ during the different state behaviour categories (Kruskal–Wallis rank-sum test, Chi squared = 2, df = 3, *p* = 0.6, Figure 4).

### 3.3. Horn Temperature in Relation to Rumination

Horn temperature was not significantly predicted by time (Table 4). The estimates of time showed a random pattern of positive and negative values, indicating that horn temperature did not consistently change during the 20 min rumination bouts (Figure 5). For model estimates and confidence intervals see Appendix A. Horn temperature varied positively with increasing THI, and negatively with windspeed, whereas the latter effect was not linear, as the windspeed category (2.5; 3.1) had the largest estimate.

## 4. Discussion

This study is one of the few to investigate the possible function of cattle horns in thermoregulation. Superficial horn temperature, measured with infrared thermography, increased significantly with increasing environmental temperature, and decreased with increasing humidity and wind speed. Horn temperature significantly increased with increasing heat load index (*HLI*), a compound variable indicating the heat load experienced by a cow. High environmental temperatures have vasodilator effects on the superficial blood vessels of peripheral structures such as the horns. This reaction increases the radiated heat, aiding heat dissipation, which can be assessed with thermographic images [28]. The positive correlation of horn temperature with HLI agrees with the definition of a heat exchange organ as described by Romanovsky [23]. Therefore, our findings strongly suggest a thermoregulatory function of the horns of dairy cows.

Against expectations, however, horn temperature did not consistently change with rumination duration during the first 20 min. The results thus do not confirm the observations of the farmers, claiming an increase in horn temperature during rumination [32].

We expected horn temperature to increase with increasing relative humidity, as well as a higher relative humidity hindering heat loss via evaporation across the rest of the body [80]. The negative correlation between horn temperature and humidity might be explained by the negative correlation between humidity and air temperature (r(571) = −7, *p* < 0.001), indicating that a lower air temperature was often accompanied by higher relative humidity, and vice versa.

The negative correlation between horn temperature and wind speed was according to expectations. A higher wind speed increases the heat loss through convection and evaporation on the skin [80]. The horns will also lose more heat with higher wind speed and thus cool down.

As the thermal camera and the camcorder both were not waterproof, it was not possible to perform observations under harsh weather conditions. Using umbrellas attracted the cows’ attention, therefore observations could only be performed when it was sufficiently dry. Therefore, weather conditions during observations may have been biased towards less humid conditions. Nonetheless, observations were carried out across a wide range of temperature, humidity, and wind speed conditions, still resulting in highly significant correlations.

The eye temperatures observed in the present study are similar to those that were reported by Salles et al. [81], though the range in the present study was broader (present study: 26.0–44.0 °C, Salles et al. [81]: 35.3–37.8 °C). This difference might be due to the experimental set-up. Salles et al. [81] performed their study in a barn, whereas the majority of the measurements in the present study were taken on pasture. There might thus have been less variety in environmental factors in the study of Salles et al. [81] than in the present study. The range in eye temperature found by Martello et al. [39] lies in-between the present study and that of Salles et al. [81], with 31.7–38.9 °C. No previous studies were found that reported horn and ear (measured on the inside of the ear) temperatures in cattle.

We found dehorned cows to have higher eye temperatures than horned cows. This is an interesting observation as eye temperature is strongly correlated to the deep body temperature [82]. The data thus suggest that dehorned cows have a higher body temperature, as HLI was similar for both groups. This result, however, should be interpreted cautiously. We cannot exclude that this finding is based on the presence of horns, differences between breeds, or due to the milk yield level that was substantially higher for the dehorned cows. A higher milk yield implies a higher metabolic rate and as a result thereof, a higher body temperature. Because only six cows of varying breed and age were observed per farm, causal conclusions on differences due to breeds or housing conditions are impossible to draw. Nevertheless, cows experienced a similar climate, and the data collection methods were standardized across farms. Follow-up studies might include a larger number of farms and cows, ideally housed in the same environment, to support the correlative relations between environmental and animal traits. As this study showed strong indications for a thermoregulatory function of the horns, differences in the risk for heat stress between horned and dehorned cows should be investigated. Additionally, a possible (long-term) effect of horn removal on stress physiology and perception, thus the activation of the sympathetic nervous system, and its influence on changes in ocular superficial temperature, should be explored. Ideally, both horned and hornless cows are studied under the same controlled environmental circumstances. Baars et al. [83] investigated horned and dehorned cows housed on the same farm under low environmental temperatures (2.2, −2.5, and −6.5 °C). The authors hypothesized milk composition to differ based on higher energy demands of horned cows necessary to compensate for additional heat loss. As feed intake was not measured, definite conclusions cannot be drawn. Nevertheless, the consequences of horn removal for physiological parameters of cattle deserves further investigation [5].

Our results add on to the negative consequences of disbudding and dehorning. Next to pain and negative mental states [7,11,84], horn removal might have negative consequences for cow welfare as the animals might have more difficulties in adapting to changing environmental conditions. Especially in light of climate change, with temperatures having risen by 0.8 degrees Celsius worldwide and 1.7 degrees Celsius in the Netherlands alone since 1900 [85], this is of high concern. Furthermore, there will be more periods with extreme temperatures, thus increasing the risk of heat stress [86]. Given the popularity of breeding polled cattle, i.e., breeds that lack the genetic disposition to grow horns, these breeds should be studied as well [87].

Safety for other cows is mentioned as one of the main reasons for horn removal, although no data on injury occurrence are available. It is important to note that Wythes et al. [88] found less bruising when cows were kept in loose housing conditions. Giving cows more space and options to move away from dominant cows reduces the risk of being injured by horned herd mates. This approach also complies with the ideas presented by Nordquist et al. [2], namely, that it is better to adapt the environment to the animal instead of the animal to its environment. These aspects, together with the results of the present study, suggest that farmers should keep horned animals to prevent the potential thermoregulatory alterations that the animals might experience.

## 5. Conclusions

The results of this study strongly indicate that cow horns have a function in thermoregulation. Horns having a functional physiological role should also be taken into consideration by farmers when deciding to disbud or dehorn their animals. For a specific function of horns during ruminating, no support was found. The present study corroborates the need for further research into the importance of horns for the welfare of cattle.

## Figures and Tables

**Figure 1 animals-13-00500-f001:**
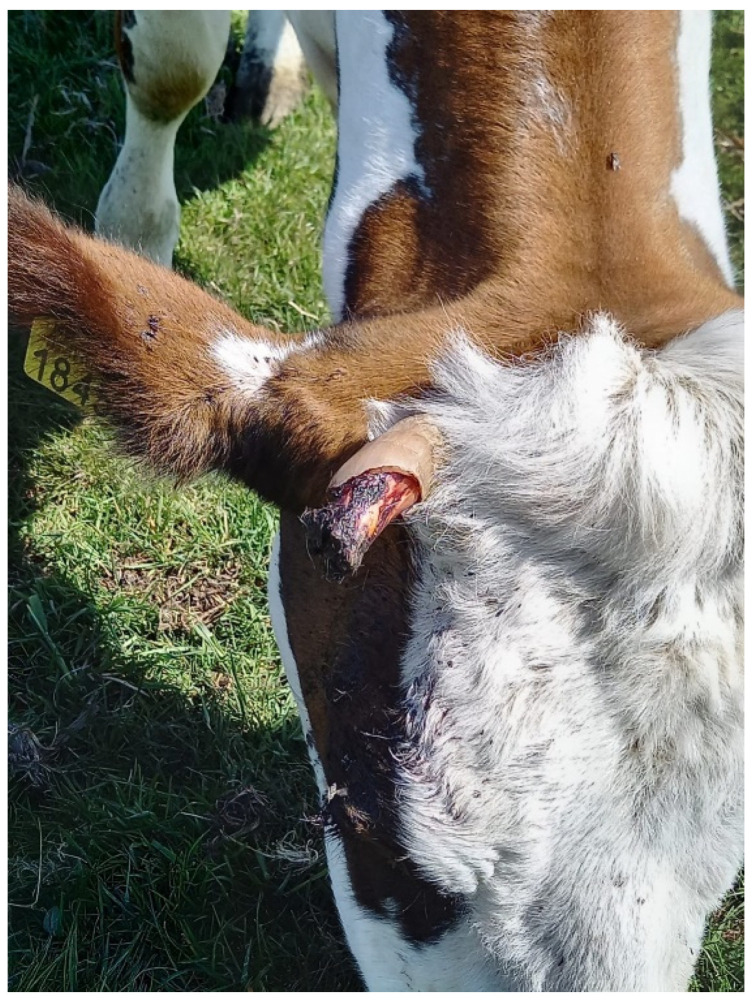
A cow at one of the farms with a broken horn. The bony core can be clearly distinguished from the keratin sheath, and it is visible that the bone contains blood vessels. (^©^Lara de Keijzer).

**Figure 2 animals-13-00500-f002:**
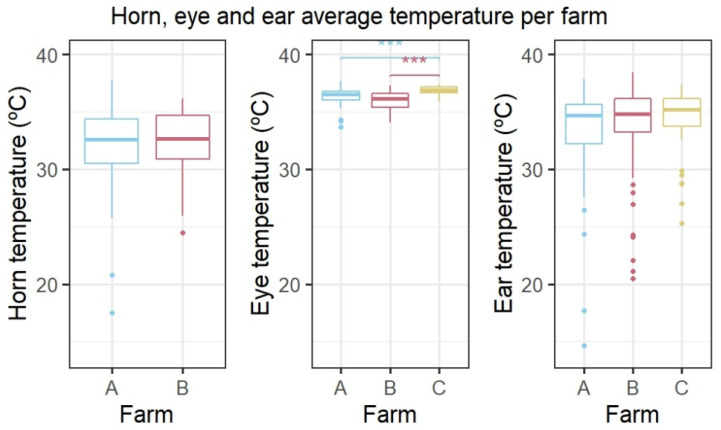
Horn, eye, and ear temperature per farm. (**Left**): horn temperature per farm. (**Middle**): eye temperature per farm. (**Right**): ear temperature per farm. Farms A and B had horned cows and farm C dehorned cows, so no difference in horn temperature could be calculated. Data were averaged per round. *** *p*-value < 0.001.

**Figure 3 animals-13-00500-f003:**
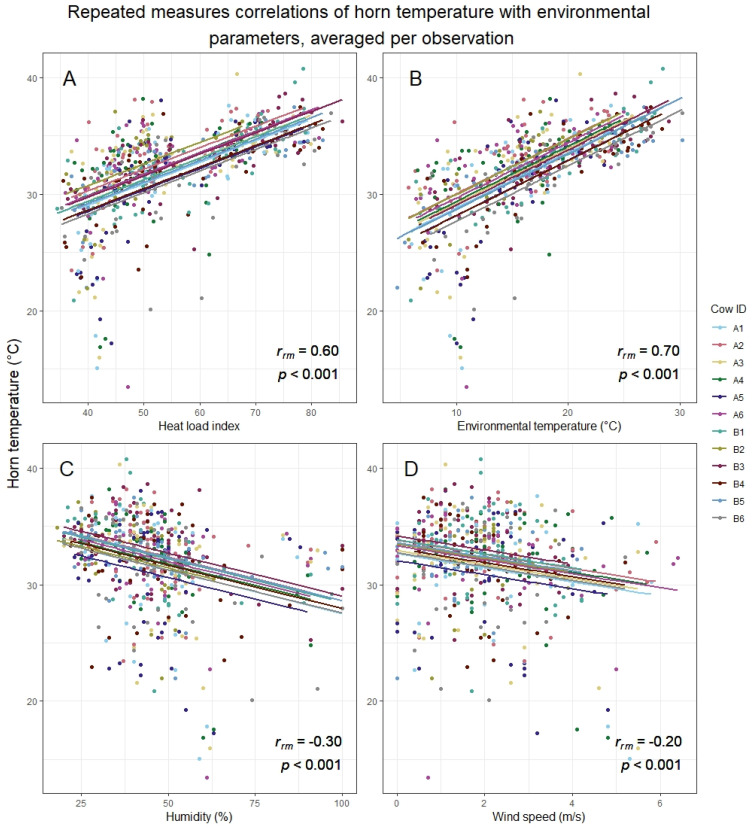
The repeated measures correlation (RMC) plots of horn temperature against heat load index (HLI), environmental temperature, humidity, and wind speed. For these plots, the horn temperatures were averaged per observation, resulting in 50 data points per cow. (**A**) RMC for horn temperature and HLI. (**B**) RMC for horn temperature and environmental temperature. (**C**) RMC for horn temperature and humidity. (**D**) RMC for horn temperature and wind speed. It should be noted that the heat load index combines air temperature, humidity, wind speed, and solar radiation. Therefore, the top left plot represents the influence of the parameters of the other three plots combined.

**Figure 4 animals-13-00500-f004:**
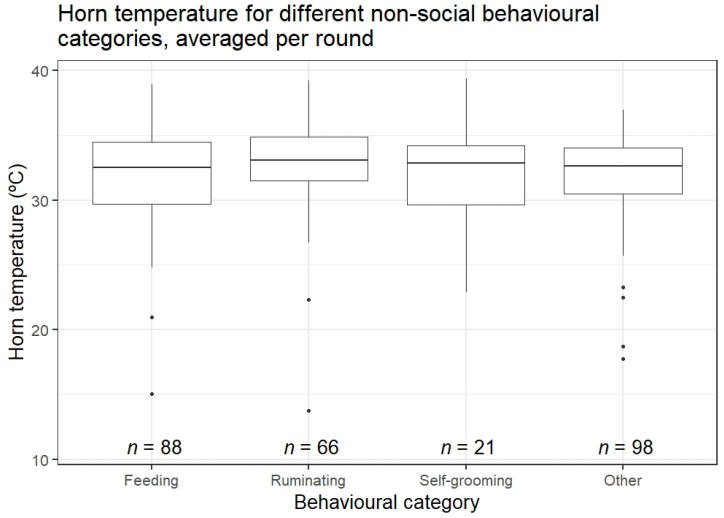
Horn temperature (averaged per round) during the different state behaviour categories.

**Figure 5 animals-13-00500-f005:**
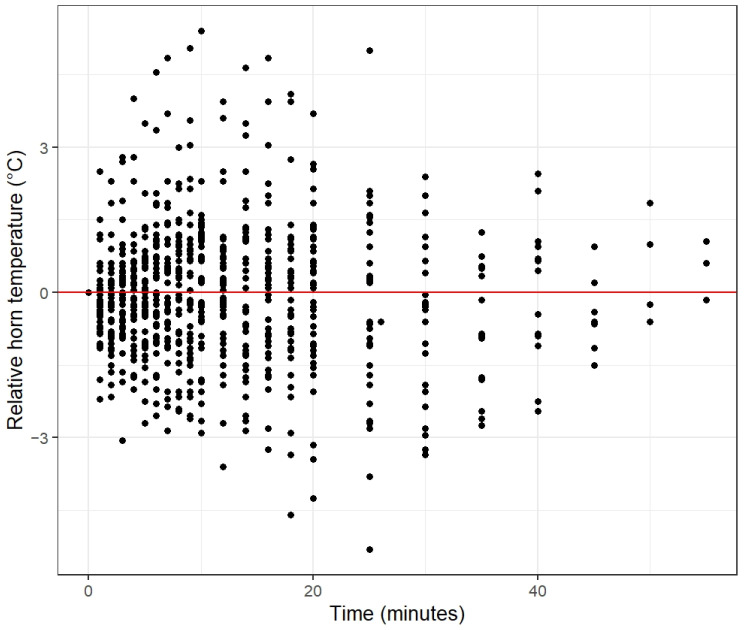
The relative horn temperature of ruminating cows (*n* = 42) plotted against the ruminating duration. All temperature values are standardized for the starting temperature. For each cow, the starting temperature was set to zero and at every subsequent time moment the starting temperature was subtracted from the horn temperature at that specific time moment, resulting in either a positive or a negative value.

**Table 1 animals-13-00500-t001:** A summary of the number of days, rounds, observations and minutes the cows were observed for data collection during the first part of the study (horn temperature in relation to behaviour and environment).

Days of Observation	Rounds	Observations	Minutes Observed
25	100 (50 each farm) Max 5 rounds per observation day	2 (farms) × 50 (rounds) × 6 (cows) = 600	50 (rounds) × 5 (minutes observed per cow per round) = 250 min per cow = 4 h 10 m

**Table 2 animals-13-00500-t002:** Ethogram of the analysed duration behaviours recorded during the first part of the study (horn temperature in relation to behaviour and environment). Videos were scored for state/duration behaviours using the continuous focal sampling method.

Category	Behaviour	Description
Non-social	Feeding	Cow is feeding on grass or other fodder; bouts of ≤10 s are ignored and disruptions of ≤10 s are allowed.
	Self-grooming	Rubbing parts of the body or head against other body parts, or licking body parts [47].
	Ruminating	Regurgitation, chewing, and swallowing of previously eaten food; bouts of ≤10 s are ignored and disruptions ≤10 s are allowed [47].
	Other	Other non-social behaviours, e.g., drinking, being alert, being milked.

**Table 3 animals-13-00500-t003:** A summary of the descriptive statistics for environmental temperature, humidity, wind speed, HLI, horn temperature, eye temperature, and ear temperature *.

Parameter	Environmental Temperature (°C)	Humidity (%)	Wind Speed (m/s)	HLI	Horn Temperature (°C)	Eye Temperature (°C)	Ear Temperature (°C)
Mean	16.1	49	1.7	55.1	32	36.4	33.7
Min	4.7	18	0	34	12	26	9
Max	30.2	100	6.4	89	54	45	55

* Note: in FLIR Research IR, only the maximum temperature of a region of interest (ROI) was extracted and used for analyses. The mean, min, and max that are given here for the horn, eye, and ear are therefore the mean, min, and max of the maximum temperature of the ROI.

**Table 4 animals-13-00500-t004:** Model fit (AIC) and *p*-values of the linear mixed-effects model testing horn temperature during the 20 min rumination bouts (*n* = 41 cows).

	Df	AIC	*p*-Value
<none>		1949.7	
Time	15	1931.7	0.68
Farm	1	1948.5	0.38
Windspeed	3	1954.5	0.01
THI	4	1978.5	0.001

## Data Availability

The data presented in this study are available in the Appendix A.

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
