# Peer review of "Evaluation of the Thermal Response of the Horns in Dairy Cattle"

_animals, 2023, doi:10.3390/ani13030500_

Round 1
Reviewer 1 Report
Authors described the horn temperature fluctuation and posible role on thermoregulation. My main issue about this study is its experimental design: I find a low sample size with different breeds. Moreover, they could compare the temperature fluctuation in horned and dehorned cows, which could enhance clarity in this knowledge gap, but maybe there were no dehorned cows in that farms...
Some other issues:
- ln 16. change noted by registered.
- ln 23. avoid double spaces
- ln 25. I would avoid using "anecdotes" word
- ln 35. just one point "."
- ln 96. avoid double spaces
- ln 113. change several by three
- ln 119. relevant literature: include references
- ln 123. why did you not use these data??
- ln 130. important gap... sample size low and 3 in dry period
- ln 138. I guess they were all Holstein, are not they? Later you explain, but breed could interefere on horn physiology
- ln 153, here you use . to separate thousands, but previously (ln 149 and 150) you used nothing. Please use ,
- ln 161. you already said this
- M&M: different breeds in the three farms, no breed standarization, and low sample size per each breed.
- Results: I would rather tables with data than many graphics, but it is a personal opinion. I would reduce the results of ruminanting, table 5 I think could be in suppl.
- Results: I think it is important to include age, farm and breed in the model. They could interfere in the results. Is this distribution equal among the whole sample size?
Reviewer 2 Report
This is an interesting paper discussing the role of horns in the thermoregulation of dairy cattle. Worldwide, dehorning is a routine practice commonly performed without anesthetic drugs that can prevent one of the studied consequences of this practice: pain. However, beyond pain perception, farmers and veterinarians need to consider the thermoregulatory role that this structure might have in cattle, and how this can help to avoid heat stress. The findings of this article will be resourceful for further studies around this topic. I left some comments hoping they can help improve the manuscript.
Line 2: Consider modifying the title of the article for something similar to “Evaluation of the thermal response of the horns in dairy cattle”
Line 15: Please, specify that it was horn, eye, and ear superficial temperature.
Lines 15-16: I recommend mentioning the behavioral traits that were evaluated (e.g., rumination, ).
Lines 23-24: To improve this sentence, it could be mentioned that little is known about the biological function and role of horns during thermoregulatory processes in cattle.
Line 26: Mention the objective of the study.
Line 31-32: Please, include the average temperature or how many degrees it incremented, as well as the HLI.
Line 36: Consider adding the keyword “infrared thermography”
Lines 39-45: I recommend re-structuring this paragraph as follows:
“In most cattle breeds, both males and females grow horns. Disbudding and dehorning are considered routine practices in more than 80% of dairy cattle in the European Union [1,2]. Disbudding refers to the removal of the horn buds in young calves, whereas dehorning is an amputation of the horns of older calves/cows when disbudding is no longer an option [3]. Several reasons are associated with these practices. For example, horned cattle are perceived to be more aggressive and there is a higher risk of causing injuries to herd mates and handlers [4,5].”
Lines 66-68: The frontal and supraorbital arteries could also be included.
Line 68: The innervation of the horn is provided as well by the cornual and auriculopalpebral nerves.
Lines 73-74: It can be added that the horn might participate in heat exchange by preserving or dissipating heat through vasomotor changes in its superficial vessels. In this way, you can associate this physiological event with the election of infrared thermography to evaluate its role during thermoregulation.
Lines 90-93: I consider that this paragraph about pain perception and its physiological and behavioral consequences could be better placed in line 51, after the reasons why dehorning is still practiced although it might have these adverse effects.
Line 95: Please, revise the objective stated in the introduction section. It differs from the one written in the simple summary.
Line 115: If the authors had a hypothesis for the present study, please, include them in the introduction after the objective.
Line 119: Please, provide a citation of the referenced study.
Lines 125-126: I recommend including how the sample size was determined for both parts of the study.
Line 145: For Farm A, B, and C characteristics, the average environmental temperature and relative humidity of the zone need to be included.
Line 193: Were those two observers veterinarians, farmers, or students? Please, include this information to understand if the observers had knowledge about cattle behavior.
Line 229: Change “Photograph” to “Radiometric image”.
Line 232: Other parameters that might need to be mentioned are emissivity and resolution.
Line 233: Although it is mentioned that a detailed protocol of the thermal imaging is in Appendix B since infrared thermography applied to the horn can be considered as a novel or non-conventional thermal window (and the key point of the present study), I recommend including a brief description on how this thermal window was delimited.
Line 238: Mention the range of hours where the thermal images were taken, and if all of them were taken at the same hour during the trial.
Lines 366-368: Consider deleting this paragraph. It doesn´t add relevant information to the article.
Lines 402-404: Please, clearly state in the title of the Figure that these are the “Horn, eye, and ear average temperatures per farm” (if these really are the average temperatures. If not, describe if they were the max, min, etc.).
Line 444: Delete “Tables may have a footer”
Lines 463-464: I strongly suggest adding a physiological explanation to why an increase in horn superficial temperature and its positive correlation to HLI suggest a thermoregulatory function. It can be added that high environmental temperatures have vasodilator effects on the superficial blood vessel of peripheral structures such as the horns. This reaction increases the radiated heat, which can be assessed with thermal imaging and, therefore, we can suggest the thermoregulatory role of horns in dairy cattle (Revise https://doi.org/10.1016/j.livsci.2015.10.022 and https://doi.org/10.3390/ani11061733
Line 495: This is an interesting finding where the discussion could also mention the role of pain perception or stress in the physiological (and thermal) response of dairy cattle. Although it is mentioned in lines 516-517, a discussion around the activation of the sympathetic nervous system and its influence on the changes in ocular superficial temperature could improve this section.
Lines 525-531: I would recommend mentioning that this could be a suggestion for farmers to keep horned animals and prevent the potential thermoregulatory alterations that the animals might have, according to the results of the present study. When leaving this paragraph without mentioning this, it seems a little out of place since the study did not evaluate the frequency of bruising or housing conditions.
Reference list: Amend references according to the journal’s instruction for authors (e.g., journal name in italics, year in bold letter, volume in italics, provide doi, etc.).
Round 2
Reviewer 1 Report
Authors accomplished my minor comments, but the main issue still unsolved: My main issue about this study is its experimental design: I find a low sample size with different breeds
Author Response
Response: We regret not having sufficiently addressed the reviewer’s concern.
The sample size for the correlation between horn temperature and environmental parameter is 12 cows. Of each of the cows, approximately 500 IRT images were taken. These were averaged per observation, meaning that the accuracy and reliability of the measurements is increased, resulting in 50 data points per cow. We added this information to the caption of figure 5. By calculating repeated measures correlations, we take the dependance of the data points into account. The statistical significance of the correlations emphasizes that horns have a thermoregulatory function despite breed (or horn size etc.). The degree of heat dissipation may vary between cow breeds, or with differently shaped horns, nevertheless, we find correlations across a range of environmental parameter in all cows (positive in the case of HLI and temperature, negative in case of humidity and wind speed).
To investigate changes within cows during rumination, we followed 11 and 31 cows, respectively, which are sufficient sample size for analysis. The model did not show an effect of farm (i.e. breed).
We agree that we cannot make causal inferences on the differences in ear and eye temperature between de/horned cows. Nevertheless, we strongly believe that this result, while potentially spurious in our study, may spark future research into the function of horns, and should therefore be included in the manuscript.
The discussion emphasizes the drawback more strongly now, and we added a sentence to the abstract and summary, again, urging cautious interpretation of the result and drawing attention to the low sample size.
Simple summary L 20: Dehorned cows had higher eye temperatures than horned cows, though this result may not be reliable due to the low sample size and experimental setup.
Abstract L 36: Dehorned cows had higher eye temperatures than horned cows, though this result should be interpreted with caution as the low sample size and experimental setup prevent casual conclusions.
Discussion L 482 ff: The data thus suggest that dehorned cows have a higher body temperature, as HLI was similar for both groups. This result, however, should be interpreted cautiously. We cannot exclude that this finding is based on the presence of horns, differences between breeds, or due to the milk yield level that was substantially higher for the dehorned cows. A higher milk yield implies a higher metabolic rate and as a result thereof, a higher body temperature. Because only six cows of varying breed and age were observed per farm, casual conclusions on differences due to breeds or housing conditions are impossible to draw. Nevertheless, cows experienced a similar climate, and the data collection methods were standardized across farms. Follow-up studies might include a larger number of farms and cows, ideally housed in the same environment, to support the correlative relations between environmental and animal traits.
Reviewer 2 Report
The authors have responded satisfactorily to all my comments.
I have no additional suggestions.
In my opinion the article should be published
Author Response
We are glad to have sufficiently addressed the feedback and for the recommendation to accept our manuscript. Thank you for your time and the constructive comments which helped improve the manuscript.
Round 3
Reviewer 1 Report
Authros accomplished my suggestions.
Even the sample size is not strong enough, I think this study contributes positively to the Academia, but further studies are needed.